# PEG–Lipid–PLGA Hybrid Particles for Targeted Delivery of Anti-Inflammatory Drugs

**DOI:** 10.3390/pharmaceutics16020187

**Published:** 2024-01-28

**Authors:** Jana Ismail, Lea C. Klepsch, Philipp Dahlke, Ekaterina Tsarenko, Antje Vollrath, David Pretzel, Paul M. Jordan, Kourosh Rezaei, Justyna A. Czaplewska, Steffi Stumpf, Baerbel Beringer-Siemers, Ivo Nischang, Stephanie Hoeppener, Oliver Werz, Ulrich S. Schubert

**Affiliations:** 1Laboratory of Organic and Macromolecular Chemistry (IOMC), Friedrich Schiller University Jena, Humboldtstraße 10, 07743 Jena, Germany; jana.ismail@uni-jena.de (J.I.); lea.klepsch@uni-jena.de (L.C.K.); ekaterina.tsarenko@uni-jena.de (E.T.); antje.vollrath@uni-jena.de (A.V.); david.pretzel@uni-jena.de (D.P.); kourosh.rezaei@uni-jena.de (K.R.); justyna.czaplewska@uni-jena.de (J.A.C.); steffi.stumpf@uni-jena.de (S.S.); baerbel.beringer-siemers@uni-jena.de (B.B.-S.); ivo.nischang@uni-jena.de (I.N.); s.hoeppener@uni-jena.de (S.H.); 2Department of Pharmaceutical and Medicinal Chemistry, Institute of Pharmacy, Friedrich Schiller University Jena, Philosophenweg 14, 07743 Jena, Germany; philipp.dahlke@uni-jena.de (P.D.); paul.jordan@uni-jena.de (P.M.J.); oliver.werz@uni-jena.de (O.W.); 3Jena Center for Soft Matter (JCSM), Friedrich Schiller University Jena, Philosophenweg 7, 07743 Jena, Germany; 4Helmholtz Institute for Polymers in Energy Applications Jena (HIPOLE Jena), Lessingstraße 12-14, 07743 Jena, Germany; 5Helmholtz-Zentrum Berlin für Materialien und Energie GmbH (HZB), Hahn-Meitner-Platz 1, 14109 Berlin, Germany

**Keywords:** hybrid nanoparticles, targeted drug delivery, BRP-201, anti-inflammatory, monocyte-derived macrophages, 5-LOX inhibition, high-performance liquid chromatography (HPLC), cryo-transmission electron microscopy (cryo-TEM), scanning electron microscopy (SEM), automated SEM image analysis, degradation

## Abstract

Hybrid nanoparticles (HNPs) were designed by combining a PLGA core with a lipid shell that incorporated PEG–Lipid conjugates with various functionalities (-RGD, -cRGD, -NH_2_, and -COOH) to create targeted drug delivery systems. Loaded with a neutral lipid orange dye, the HNPs were extensively characterized using various techniques and investigated for their uptake in human monocyte-derived macrophages (MDMs) using FC and CLSM. Moreover, the best-performing HNPs (i.e., HNP-COOH and HNP-RGD as well as HNP-RGD/COOH mixed) were loaded with the anti-inflammatory drug BRP-201 and prepared in two size ranges (d_H_ ~140 nm and d_H_ ~250 nm). The HNPs were examined further for their stability, degradation, MDM uptake, and drug delivery efficiency by studying the inhibition of 5-lipoxygenase (5-LOX) product formation, whereby HNP-COOH and HNP-RGD both exhibited superior uptake, and the HNP-COOH/RGD (2:1) displayed the highest inhibition.

## 1. Introduction

The application of drugs is a delicate trade-off between their therapeutic and side effects because conventional medications distribute drugs systemically throughout the whole body and thus affect both healthy and diseased cells and organs [1]. To tackle this challenge, extensive research has been conducted over the past decades to develop targeted drug delivery systems (DDSs) that enable a directed transport of active compounds to the desired site of action, rendering the need for high doses obsolete and therefore reducing side effects [2]. The application of DDSs is particularly required for lipophilic drugs that are poorly water soluble and need protection from premature degradation and metabolism [3]. Representatives of such DDSs are hydrogels [4], nanofibers [5], nanocomposites [6], and lipid- and polymer-based nanoparticles [7,8], amongst others. Polymer-based carriers offer tunable stability and biodegradability through a variation in polymer properties. A famous example is the polyester poly(D,L-lactic-*co*-glycolic acid) (PLGA), which is U.S. Food and Drug Administration (FDA) approved and has long been used as a biomaterial [9,10]. However, in terms of their biocompatibility and safety profile, lipid-based DDSs, e.g., in the form of liposomes, offer superior properties because they are basically analogues of biological membranes [11]. They are, however, rapidly cleared from the bloodstream and often suffer from poor pharmacokinetics [12]. Moreover, liposomes lack structural integrity, making them susceptible to drug leakage and long-term stability issues [11]. Hence, to take advantage of the unique properties of the polymer- and lipid-based systems and overcome their disadvantages, the combination of both led to the creation of so-called lipid–polymer hybrid nanoparticles (HNPs) [13]. HNPs often consist of a core–shell structure with an inner hydrophobic polymeric core surrounded by a lipid layer on the outside [14]. The lipid layer can consist of lecithin or cholesterol and an additional lipid, e.g., DSPE (1,2-distearoyl-sn-glycero-3-phosphoethanolamine) or DOPE (1,2-dioleoyl-sn-glycero-3-phosphoethanol-amine), which are conjugated with polyethylene glycol (PEG). The function of the lipid layer is to ensure biocompatibility and stability, while the functional PEGylated lipids additionally improve the water solubility of the particles, reducing the aggregation tendency through steric stabilization [11,13,15]. The PEG shell serves as a stealth coating that prevents opsonization, resulting in a prolonged circulation time of DDS in the bloodstream [16,17]. In addition, PEG–lipid conjugates can carry a functional group at the end of the PEG chain, which can also be utilized for a straightforward introduction of desired targeting units. Given the promising properties of combining the polymer- and lipid-based systems, it is not surprising that HNPs have been studied in more detail in the past years in terms of their formulation conditions, structure, and carrier capabilities. The first publication of the formulation of HNPs as a DDS was published in 2002 for the delivery of luteinizing hormone-releasing hormone (LHRH) [18]. Following that, in 2008, the group of Robert Langer and Omid Farokhzad formulated HNPs for the delivery of docetaxel, a widely used anticancer drug [19]. Since then, HNPs have been described not only to deliver drugs but also genes [20] and dyes for imaging [21].

The present study aimed to employ HNPs for the targeted transport to human monocyte-derived macrophages (MDMs) and neutrophils for an enhanced delivery of the anti-inflammatory drug BRP-201 (5-{1-[(2-chlorophenyl)methyl]-2-{1-[4-(2-methylpropyl)-phenyl]ethyl}-1H-benzimidazole-5-yl}-2,3-dihydro-1,3,4-oxadiazole-2-thione), which inhibits the 5-lipoxygenase (5-LOX) pathway (the structure is shown in Appendix A; Appendix A are indicated in the following by the letter S in the figure captions). PEG–Lipid–PLGA-based HNPs were prepared using DSPE–PEG–Lipids with different functional groups, i.e., -NH_2_ and -COOH, and active targeting moieties for the integrin receptor found on various cells including macrophages, i.e., arginylglycylaspartic acid (RGD) and cyclic arginylglycylaspartic acid (cRGD) [22], resulting in HNP-NH_2_, HNP-COOH, HNP-RGD, and HNP-cRGD, respectively. At first, HNPs were loaded with a fluorescent dye as cargo to track their cellular uptake depending on the surface moiety applied. Subsequently, the best-performing HNPs were selected and loaded additionally with the anti-inflammatory drug BRP-201, whereby HNPs with mixtures of functionalized PEG–Lipids (i.e., DSPE-PEG-COOH/DSPE-PEG-RGD at ratios of 1:1 and 2:1 (*w*/*w*)) also were prepared. The cellular uptake of HNPs was investigated by employing these systems to explore the influence of the amount of active targeting present on the particle surface. Moreover, the size of the HNPs was varied to evaluate the passive targeting effect of the particle size [23]. The dual encapsulation of the dye and the drug into the HNPs was carried out to investigate the cellular uptake by MDMs and the delivery of BRP-201 to both MDMs and neutrophils in parallel. Human MDMs were selected because they play a crucial role in initiating, sustaining, and resolving inflammatory processes [24], and neutrophils were chosen because they are highly abundant innate immune cells in the blood that provide a strong capacity to generate pro-inflammatory 5-LOX products [25]. The drug BRP-201 acts as a 5-LOX activating protein (FLAP) inhibitor with an IC_50_ value of 50 nM [26]. It belongs to a new class of anti-inflammatory drugs with superior properties compared to the current anti-inflammatory therapies available [26,27,28]. However, BRP-201 is a lipophilic drug with a very low water solubility and hence low bioavailability, which is why it relies on an encapsulation into a suitable DDS [29].

## 2. Materials and Methods

### 2.1. Materials

The acid-terminated PLGA copolymer (Resomer RG 502 H) with a ratio of LA:GA of 50:50 and a molar mass of 7000 to 17,000 g mol^−1^ was purchased form Evonik (Darmstadt, Germany). PEG-PLGA (PEG average M_n_ = 5000 g mol^−1^, PLGA M_n_ = 5000 g mol^−1^ with a feed ratio of LA:GA of 50:50) was sourced from Sigma Aldrich (Darmstadt, Germany). The functional lipids, including DSPE-PEG(2000)-Amine (Merck, Darmstadt, Germany), DSPE-PEG(2000)-Carboxylic acid (Merck, Darmstadt, Germany), DSPE-PEG(2000)-RGD (Biopharma, San Diego, CA, USA), and DSPE-PEG(2000)-cRGD (Biopharma, San Diego, CA, USA), were obtained as well. Lecithin (EMD-Millipore, Darmstadt, Germany) was used as a helper lipid, and the NLO dye (Dyomics, Jena, Germany) was incorporated into the HNPs. BRP-201 was synthesized according to a published procedure [30]. Dimethyl sulfoxide (DMSO), acetonitrile (CH_3_CN), ammonium acetate (≥97%), partially hydrolyzed PVA (Mowiol 4-88), and 15 mL Amicon^®^ filters (100,000 g mol^−1^ MWCO) were obtained from Sigma Aldrich (Darmstadt, Germany). LC-MS grade water, acetonitrile (CH_3_CN), and methanol (CH_3_OH) were purchased from VWR (Darmstadt, Germany). Acetic acid was acquired from Merck KGaA (Darmstadt, Germany). PrestoBlue™ Cell Viability Reagent, Hoechst 33342, LysoTracker™ Green DND-26, and CellMask™ Deep Red membrane stain (CMDR) were procured from Thermo Fisher Scientific (Waltham, MA, USA). FITC Mouse Anti-Human CD14 and FITC Mouse IgG2a, κ Isotype Control were obtained from BD Biosciences (Franklin Lakes, NJ, USA).

### 2.2. Methods

#### 2.2.1. Formulation of HNPs and NPs

The HNPs were synthesized using a single-step nanoprecipitation technique (Appendix A). The procedure was adopted with modifications from J. M. Chan [31]. To begin, PLGA was dissolved in acetonitrile (CH_3_CN) at concentrations of 2.5 mg mL^−1^ or 25 mg mL^−1^ (Appendix A). The NLO dye and BRP-201 drug were dissolved in DMSO at concentrations of 1 mg mL^−1^ and 10 mg mL^−1^, respectively, and added to the polymer solution at 0.1 wt% (NLO) and 3 wt% (BRP-201) relative to the PLGA (Appendix A). Lecithin and DSPE-PEG with various functionalities (-NH_2_, -COOH, -RGD, and -cRGD; Appendix A) were dissolved in an aqueous solution containing 4 wt% ethanol (accounting for 15 wt% of the PLGA weight) and heated to approximately 65 °C. The organic polymer solution was subsequently gently stirred and added to the preheated lipid aqueous solution using a syringe pump (2 mL min^−1^) followed by vortexing for two min. Polyvinyl alcohol (PVA) was added afterwards at a range of 5 wt% to 25 wt% relative to the polymer mass. The mixture was stirred continuously at 800 rpm at room temperature under exclusion of light, allowing the organic solvent to evaporate overnight. Afterwards, the pure dispersion was analyzed using dynamic light scattering (DLS). After the organic solvent evaporated, the HNPs were purified using Amicon^®^ Ultra-4 filters (100 kDa, Merck, Sigma Aldrich, Darmstadt, Germany) and a 5804 R centrifuge (Eppendorf, Germany) at 4500× g for 20 min at 20 °C. The supernatant was separated, and the small particles were washed with 10 mL of pure water, while the large particles were washed with 4 mL of pure water. The volume of water was restricted by the volume of the Amicon^®^ filters used. Both fractions were centrifuged again at 5000 rpm for 15 min at 20 °C. After separating the supernatant, 2 mL of milliQ water were added to resuspend the particles, and the dispersions were stored overnight at 4 °C. On the following day, the size and zeta potential of the particles were measured. Subsequently, the HNPs and PEG-PLGA were sterile-filtered through a 0.8 µm cellulose acetate filter. To determine the concentration after filtration, two aliquots of 250 µL were freeze-dried using a Lyophilizer Christ Alpha 2-4 LD plus (Martin Christ, Osterode am Harz, Germany), and the sample mass was determined using a MYA 11.4Y microbalance (Radwag Waagen, Hilden, Germany). All dispersions were brought to an equal concentration with milliQ under a sterile bench before being used for cellular uptake studies.

#### 2.2.2. Dynamic and Electrophoretic Light Scattering (DLS and ELS)

The particles were characterized using dynamic light scattering (DLS) and electrophoretic light scattering (ELS) measurements conducted on a Zetasizer Ultra (Malvern Panalytical GmbH, Malvern, UK) with a laser wavelength of λ = 633 nm. For determination of size, polydispersity (PDI), and zeta potential, UV cuvettes made of polystyrene (Brand, Wertheim, Germany) or DTS1070 capillary cuvettes (polycarbonate, Malvern Panalytical GmbH, Kassel, Germany) were utilized. The measurements were conducted at a back-scattering angle of 174.7° and a temperature of 25 °C. The hydrodynamic diameter (d_H_) of the HNPs was determined based on the size distribution according to intensity. Measurements were performed with a dilution of 1:10 or 1:100 using pure water, taking five measurements with 15 runs each for d_H_ and PDI and three measurements each for zeta potential determination. In addition, the zeta potential was evaluated with a 1:100 dilution in 0.01 M of sodium chloride. To assess the stability of the HNPs in buffer, the dispersion was measured in a 1:1 or 1:10 dilution with the appropriate buffer, and the evaluation was conducted following the same procedure. For the degradation studies, the particles were incubated with a 1 mg mL^−1^ proteinase K solution in water at a mass ratio of 1:2, and the mean count rate in kilocounts per second (kcps) was monitored over a period of 12 h while maintaining a constant attenuator setting of six.

#### 2.2.3. UV–Vis Spectroscopy

To determine the loading of the drug and dye, UV–Vis measurements were carried out using the Infinite M200 Pro plate reader (Tecan Group, Männedorf, Switzerland). Lyophilized HNP samples were dissolved in DMSO and stirred for 30 min. Subsequently, 100 μL of undiluted and diluted solutions were measured in a Hellma black quartz plate with 96 wells. For the BRP-201 drug, absorbance measurements were taken at λ_ex_ = 316 nm using 3 × 3 multiple reads per well and a well border of 2000 μm. For fluorescence measurements of the NLO dye, 3 × 3 multiple reads per well and a well border of 1700 μm were used with excitation at λ_ex_ = 555 nm and emission at λ_em_ = 592 nm. The loading capacity (LC) was calculated by dividing the mass of the drug recovered by the mass of particle recovered multiplied by a factor of 100 [29].

#### 2.2.4. Scanning Electron Microscopy (SEM)

Imaging of the particles was performed using a Sigma VP Field Emission Scanning Electron Microscope (Carl-Zeiss, Jena, Germany) equipped with an Inlens detector and operated at an accelerating voltage of 5 to 6 kV. Prior to imaging, the samples were drop-casted on a mica plate and coated with a 4 nm platinum layer using a CCU-010 HV sputter (Safematic, Zizers, Switzerland).

#### 2.2.5. Analysis of the Particle Size Distributions in SEM Images

The analysis of the particle size distributions in the SEM images were realized using openCV in python [32]. By following the algorithm outlined in the Appendix A, the analysis process was automated, leading to the valuable insights regarding the size distribution of particles within the SEM images. The algorithm used involved a series of preprocessing steps and an iterative approach to detect particles until no more were found in the image.

#### 2.2.6. Cryo-Transmission Electron Microscopy (Cryo-TEM)

Transmission electron microscopy was performed on a FEI Tecnai G^2^ 20 system. First, 15 µL of the sample solution was blotted onto a precleaned carbon coated grid (Quantifoil, Großlöbichau, Germany). Cleaning was performed with a Pelco Easy Glow™ glow discharge system for 20 s. After blotting, the grids were stained with UranyLess EM contrasting solution for 5 min (Science Services, München, Gemany). Images were acquired with an acceleration voltage of 200 kV utilizing an OSIS Megaview CCD camera (Olympus Soft Imaging Solutions, Tokyo, Japan). 

#### 2.2.7. Size-Exclusion Chromatography (SEC)

SEC measurements were conducted to confirm the presence of all the starting material in the final HNP formulations using an Agilent 1200 series system equipped with a G1310A pump, a G1315D diode array detector (DAD), a G1362A refractive index (RI) detector, and PSS GRAM 30 Å/1000 Å (10 µm particle size) columns in series. The SEC analysis was performed at a temperature of 40 °C while employing *N*,*N*-dimethylacetamide (DMAc) with 2.1 g L^−1^ LiCl as the eluent and utilizing a flow rate of 1 mL min^−1^. The system was calibrated using polystyrene (PS) standards within a molar mass range of 374 to 1,040,000 g mol^−1^.

#### 2.2.8. High-Performance Liquid Chromatography (HPLC)

The composition of formulated dual-loaded HNPs of smaller size was monitored via HPLC using an UltiMate 3000 Dionex UHPLC chromatographic system from Thermo Fisher Scientific (USA). The column oven temperature was set to 35 °C and the autosampler temperature to 17 °C. The flow rate was 0.75 mL min^−1^, and 5 µL of the sample was injected. Elutions were monitored by three different detectors: a charged aerosol detector (Corona^TM^ Veo^TM^ RS CAD), a diode array detector (DAD), and a fluorescence detector (FLD). The CAD was used as a universal detector for monitoring the composition of the NPs and HNPs. The data were collected at a 5 Hz acquisition frequency with the nebulizer tempered at 45 °C. Simultaneously, the detection of BRP-201 was enabled by the DAD operating at the absorbance maximum of the drug (λ_abs_ = 312 nm), and the NLO dye was detected via FLD (λ_ex_ = 555 nm, λ_em_ = 592 nm). As a stationary phase, a reversed-phase, silica-based monolithic column was utilized. The Chromolith^®^ Performance RP-8 endcapped column (100 × 4.6 mm) from Merck KGaA (Darmstadt, Germany) was operated with a ternary mobile phase gradient composition. The mobile phase consisted of (A) CH_3_CN, (B) water with 10 mM ammonium acetate with the pH of 5.5 adjusted with acetic acid, and (C) CH_3_OH with 10 mM ammonium acetate. The final elution program included an isocratic hold at starting conditions (A/B, 55/45, %, *v*/*v*) followed by a linear gradient to higher CH_3_OH content in the mobile phase (Appendix A). Data were processed by using the Thermo Scientific™ Dionex™ Chromeleon™ 7.2 (SR5) Chromatography Data System software.

For the HPLC analysis, lyophilized aliquots of the formulations were dissolved in 50/25/25 (%, *v*/*v*/*v*) DMSO/CH_3_CN/CH_3_OH and sonicated for 5 min at room temperature. DSPE-PEG-COOH, DSPE-PEG-RGD, and lecithin standards were dissolved at a concentration of 1 mg mL^−1^ in CH_3_OH. PVA (30 mg mL^−1^ in water) was diluted up to 1.5 mg mL^−1^ with DMSO/CH_3_CN/CH_3_OH 50/25/25 (%, *v*/*v*/*v*). For calibration, BRP-201 and PLGA standards were dissolved in DMSO to obtain a stock solution with a concentration of 1000 µg mL^−1^ and 20 mg mL^−1^, respectively. The stock solutions were diluted to two series in terms of concentration: 40, 30, 25, 15, 10, and 5 µg mL^−1^ for BRP-201 and 5.0, 2.5, 1.5, 1.0, 0.5, and 0.1 mg mL^−1^ for PLGA. The final solvent composition in all analyses was DMSO/CH_3_CN/CH_3_OH at 50/25/25 (%, *v*/*v*/*v*). Prior to the analyses, all samples were filtered through a hydrophobic polytetrafluoroethylene (PTFE) filter (AppliChrom GmbH, Oranienburg, Germany) with a 0.45 µm pore size. 

#### 2.2.9. Isolation of Neutrophils and Macrophages

Leukocyte concentrates acquired from freshly withdrawn blood (containing 16 I.E. heparin/mL blood) of healthy adult male and female volunteers (18 to 65 years old) were provided by the Department of Transfusion Medicine at the University Hospital of Jena, Germany. The experimental procedures were approved by the local ethical committee of the University Hospital of Jena (Germany; approval no. 5050–01/17) and were performed in accordance with the respective guidelines and regulations. Volunteers agreed to the usage via written consent. According to previously published procedures [33], neutrophils and peripheral blood mononuclear cells (PBMCs) were isolated via density gradient centrifugation using a lymphocyte separation medium (C-44010, PromoCell, Heidelberg, Germany) after sedimentation of erythrocytes using dextran. PBMCs were seeded in phosphate-buffered saline (PBS) pH 7.4 containing CaCl_2_ and MgCl_2_ (Sigma-Aldrich, Darmstadt, Germany) in cell culture flasks (Greiner Bio-One, Frickenhausen, Germany) for isolation of monocytes. The medium was discarded and replaced with RPMI 1640 (Thermo Fisher Scientific, Schwerte, Germany) containing heat-inactivated fetal calf serum (FCS, 10% *v*/*v*), penicillin (100 U/mL), streptomycin (100 μg mL^−1^), and L-glutamine (2 mmol L^−1^) after 1 h at 37 °C and 5% CO_2_ for monocyte adherence.

#### 2.2.10. Differentiation and Polarization of Monocytes to Monocyte-Derived Macrophages

Differentiation of monocytes to macrophages and subsequent polarization to M_1_-like phenotypes were carried out as described previously [34]. In brief, PBMCs were incubated with 20 ng mL^−1^ GM-CSF (Cell Guidance Systems Ltd., Cambridge, UK) for six days in RPMI 1640 supplemented with 10% FCS, L-glutamine, penicillin, and streptomycin. The obtained M_0 GM-CSF_ MDMs were then treated with LPS (100 ng mL^−1^) and IFN-γ (20 ng mL^−1^; PeproTech, Hamburg, Germany) for 24 h to obtain M_1_-MDMs.

#### 2.2.11. Incubation for Lipid Mediator (LM) Formation and LM Metabololipidomics via UPLC–MS-MS

To study the effects of the particles on cellular LM formation, neutrophils (5 × 10^6^ mL^−1^) were preincubated with BRP-201-loaded NPs (PEG-PLGA NPs, HNP-COOH, HNP-RGD, and HNP-COOH/RGD (1:1; 2:1)) or PBS as control for 15 min at 37 °C. Neutrophils were treated with 2.5 µM of A23187 (Cayman, Hamburg, Germany) for 10 min to induce 5-LOX product formation. The incubation was stopped with 1 mL of ice-cold methanol containing 200 ng/mL PGB_1_ as an internal standard. The 5-LOX products (LTB_4_, trans-isomers of LTB_4_, and 5-hydroperoxyeicosatetraenoic acid (5-HETE)) were extracted via solid-phase extraction (SPE) and analyzed via reversed-phase HPLC as previously described [35]. Human monocyte-derived M_1_-like macrophages (1 × 10^6^ mL^−1^ M_1_-MDMs) were seeded in 12-well plates and preincubated for 15 min at 37 °C and 5% CO_2_. Subsequently, the cells were incubated with *S. aureus*-conditioned medium (SACM, 1%) of the 6850 strain as a stimulus for 90 min. The reaction was stopped with ice-cold methanol containing deuterated LM standards (200 nM of d8-5S-HETE, d4-LTB4, d5-LXA4, d5-RvD2, and d4-PGE2 and 10 μM of d8-AA; Cayman Chemical/Biomol GmbH, Hamburg, Germany). Samples were kept at −20 °C for one day to allow protein precipitation for at least 60 min. After centrifugation (2000× *g* at 4 °C for 10 min), 8 mL of acidified water was added (final pH = 3.5) and samples were subjected to SPE using RP-18 columns, and the LMs were analyzed via ultra-performance liquid chromatography–tandem mass spectrometry (UPLC-MS–MS) using an Acquity UPLC system (Waters, Eschborn, Germany) and a QTrap 5500 Mass Spectrometer (Sciex, Framingham, MA, USA) equipped with an electrospray ionization source exactly as described before [34].

#### 2.2.12. Determination of Cell Viability

The in vitro cytotoxicity tests were performed using a PrestoBlue™ assay following the supplier’s instructions. Macrophages (M_0_-MDMs) were seeded into 96-well plates at a density of 1 × 10^5^ cells per well in RPMI 1640 and were left to adhere for 24 h. Cells were then incubated at 37 °C for 24 h with 1.8, 18, and 180 µg mL^−1^ concentrations of each HNP sample (n = 3). Control cells were incubated with fresh media and 10% purified water. Then, the media was discarded, and 100 μL of fresh media containing 10% PrestoBlue™ was added to each well. After incubation for 3 h at 37 °C, the fluorescence was measured at λ_ex_ = 560 nm and λ_em_ = 590 nm in bottom reading mode with five flashes per scan using an Infinite M200 Pro plate reader (Tecan Group, Männedorf, Switzerland). The cytotoxicity was expressed as the percentage of the cell viability as compared with the blank control.

#### 2.2.13. Evaluation of Cellular Uptake of NLO-Containing HNPs via Confocal Laser Scanning Microscopy (CLSM)

To investigate the uptake of NLO-containing HNPs using confocal microscopy, M_0_-MDMs were seeded into 96-well black, clear-bottom plates (165305, Thermo Fisher Scientific, Waltham, MA, USA) at a density of 1 × 10^5^ cells per well in RPMI 1640 (n = 3) and were left to adhere overnight. After 24 h of incubation, the cells were treated with the HNP at 180 µg mL^−1^ for 3 h. Then, the media was discarded, and the cells were washed once with PBS. They were subsequently stained with Hanks’ Balanced Salt Solution (HBSS) phenol-red-free media containing 10 µg mL^−1^ Hoechst 33342 (Invitrogen, Carlsbad, CA, USA), 0.2 µM LysoTracker™ Green DND-26 (Invitrogen, Carlsbad, CA, USA), and CellMask™ deep red membrane stain (CMDR) (Thermo Fisher Scientific, Waltham, MA, USA) (diluted at 1:1000 of the concentrated stock solution) for 20 min and washed later three times with fresh PBS. The HBSS phenol-red-free media was then replenished, and the intracellular localization was detected with a Zeiss LSM 880 Elyra (Carl Zeiss, Oberkochen, Germany). The Hoechst 33342, LysoTracker™, NLO, and CMBR channels were excited at 405 nm, 488 nm, 561 nm, and 633 nm, respectively, and the emission was collected in the spectral intervals of 410 to 498 nm, 490 to 588 nm, 570 to 633 nm, and 661 to 759 nm, respectively. Images from an ROI of 125 × 125 µm were acquired using a 40× water-immersion objective (N.A. 1.2) with a pixel size of 80 nm. Moreover, the same experimental setup except the lysosome and cell membrane staining was followed to study the M_0_-MDMs uptake kinetics of the different HNPs (with imaging performed every five min for 1 h) as well as that of the HNP-COOH, HNP-RGD, and PEG-PLGA nanoparticles while comparing them to pure NLO (dissolved in DMSO) in PBS by M_1_-MDMs macrophages, with confocal images captured every two min for a duration of 15 min.

#### 2.2.14. Evaluation of NLO Uptake via Flow Cytometry

To investigate the uptake of NLO-containing HNPs by macrophages using flow cytometry, M_0_-MDMs were seeded into 24-well plates at 37 °C and 5% CO_2_ at a density of 5 × 10^5^ cells per well in RPMI 1640 and were left to adhere overnight. After 24 h of incubation, the cells were treated for 3 h with increasing concentrations of the HNPs (1.8, 18, and 180 µg mL^−1^). These concentrations were prepared by diluting the stock of 1.8 mg mL^−1^ to 18, 180, and 1800 µg mL^−1^, respectively, to be further diluted (1:10) in the cell media in the wells. Then, the media was discarded, and the cells were detached with PBS containing 5 mmol L^−1^ EDTA and collected into a 96-well plate. HNP uptake was determined in 10,000 cells by quantifying the NLO fluorescence using the CytoFlex LX (Beckman Coulter GmbH, Krefeld, Germany) at λ_ex_ = 561 nm with a 585/42 bandpass filter. The side-scattered and forward-scattered light signals were used to exclude debris and aggregates from the heterogeneous cell suspension, and an FITC CD14 staining was performed to discriminate target populations (see the protocol in the Appendix A). The data were analyzed using the CytExpert 2.5 software, and the correction factor derived from the HNP fluorescence was applied to the raw MFI values (n = 3). 

An identical experimental setup was implemented to study the M_1_-MDM macrophages’ uptake of smaller and larger PEG-PLGA nanoparticles, HNP-COOH, HNP-RGD, HNP-COOH/RGD (1:1), and HNP-COOH/RGD (2:1) except for a set concentration of 100 µg mL^−1^ post-incubation for 3 h.

#### 2.2.15. Statistical Analysis

The results are given as the mean ± standard error of each independent experiment, where n represents the indicated number from different donors. The statistical analysis was conducted and graphs were created by using GraphPad Prism 9 software (San Diego, CA, USA). The statistical analysis was assessed using a ratio paired *t*-test, with a *p*-value below 0.05 indicating a significant result.

## 3. Results and Discussion

### 3.1. Formulation of Dye-Loaded PEG–Lipid–PLGA HNPs

Initially, the formulation conditions of the HNPs were adapted from the study by Robert Langer and Omid Farokhzad [31] and fine-tuned using DSPE-PEG-NH_2_ and DSPE-PEG-COOH as functional lipids (Appendix A in the Appendix A). PEG with a molar mass of 2000 g mol^−1^ was utilized in the PEG–Lipid conjugates because this molar mass was already successfully applied in previous studies [36]. After testing different ratios of lecithin and the PEG–Lipids, a 1:2 ratio of lecithin to DSPE-PEG was found to be best for an efficient and reliable particle formulation (Figure 1 and Appendix A in the Appendix A). However, it was also revealed in the test formulations that a minimum amount of 5 wt% of the surfactant poly(vinyl alcohol) (PVA) was required to maintain the stability of the HNPs over four weeks (Appendix A and Appendix A in the Appendix A). The presence of all the starting compounds (polymer, helper lipid, and functional lipid) in the final HNP formulations was confirmed via SEC analysis of the purified lyophilized dispersions (Appendix A in the Appendix A). Moreover, the core–shell structure of the hybrid particles was proven via cryo-TEM analysis by utilizing negative staining to visualize the core–shell structure (Figure 1B).

After optimization of the formulation conditions, HNPs with different functional groups were prepared, whereby in addition to the DSPE-PEG-NH_2_ and DSPE-PEG-COOH lipids, two additional peptide functionalized PEG–Lipids, i.e., DSPE-PEG-RGD and DSPE-PEG-cRGD, were utilized. The final formulations are further referred to as HNP-NH_2_, HNP-COOH, HNP-RGD, and HNP-cRGD, respectively. In addition, PEG-PLGA NPs were prepared as a control system without lipids (Appendix A). The hydrophobic NLO dye was encapsulated in all particle systems for their visualization in the cellular uptake studies [37]. The final properties of the different HNPs and NPs loaded with NLO are presented in Table 1 and Appendix A. The mean size (herein displayed as d_H_) of all particles ranged between 125 nm and 175 nm with PDI values below 0.15 (Appendix A).

To confirm the DLS results and to visualize the particles, SEM measurements of the HNPs were performed, and the mean size (number-weighted value, d_N_) of the particles was calculated from the SEM images via image processing (Table 1; Appendix A). The d_N_ values of the HNPs were found to be in the range of 72 nm to 98 nm and confirmed the sizes detected via DLS. The shift to smaller mean diameters in the SEM analysis was expected and was caused by the measurement in the dried state compared to the DLS measurements, which determined the hydrodynamic diameter of the particles in suspension. Overall, the obtained particle sizes were similar for the different functions and were in a valid range for targeted drug delivery applications; it should be mentioned here that the uptake of particles by macrophages generally increases with the particle size [38]. However, in a previous study by C. He, it was reported that the uptake by macrophages can also be increased by increasing the surface charge [39]. An effective uptake with particles in the size range of 150 nm and strongly negative zeta potentials in the range of −30 mV were shown. The HNPs prepared herein also revealed strongly negative zeta potentials (ζ = −29 to −35 mV) for the HNP-COOH, HNP-RGD, and HNP-cRGD as well as moderate values between −14 mV and −20 mV for the HNP-NH_2_ and the control PEG-PLGA particles (Table 1).

The prepared HNP dispersions remained stable in water, in PBS (pH = 5.8 to 7.5), and in acetate buffer (pH = 5.0) over at least one week up to four weeks (Appendix A; Appendix A). The HNP-cRGD tended to aggregate over time, which was explained by the lower overall zeta potential (<15 mV) and the inefficient repulsion of the particles by each other [40]. The degradation in the particles was studied enzymatically using proteinase K (Appendix A) as described in a previous study [41]. With utilization of an enzyme-to-particle ratio of 2:1, the HNPs revealed fast degradation half-lives of about 10 min, and the PEG-PLGA NPs showed slightly slower degradation times of about 40 min. A study by C. Kretzer found that PLGA-based NPs exhibited degradation half-lives of around 20 days in PBS [29]. The rapid degradation in the particles shown here can be explained by the usage of an enzyme. Proteinase K is a hydrolase and therefore a catalyst for hydrolysis reactions, which accelerates the degradation of the particles [42]. In principle, the degradation behavior of the particles can be adjusted in a defined manner via the properties of the PLGA. By varying the molar mass (M_w_), the monomer ratios (LA:GA), the glass transition temperature (T_g_), or the polymer end groups, the rate of degradation can be controlled [43].

### 3.2. Cytotoxicity and Uptake Studies of Dye-Loaded Particles in M_0_-MDMs

In order to assess the safety of the formulated HNPs, experiments were conducted on M_0_-MDMs (Figure 2A). Exposing the macrophages to the HNPs provided a generalized picture of the safety of the HNPs on all macrophage lineages. The HNPs were tested at concentrations ranging from 1.8 µg mL^−1^ to 180 µg mL^−1^ over a 24 h period (Appendix A). The highest concentration was determined as the highest available HNP concentration formulated after purification, which was calculated to be 1.8 mg mL^−1^. This concentration was further diluted in media (1:10) to 180 µg mL^−1^. The results revealed that there was no observed decrease in cell viability over time, suggesting that the particles are safe and non-toxic (Figure 2A). These findings, along with previous evidence demonstrating the safety of PEG-PLGA NPs [44,45], established a similar safety profile for the HNPs. Moreover, this finding became particularly noteworthy given that the confirmed safety was specifically assessed in macrophages derived from human donors.

Prior to proceeding with the uptake studies, MDMs were stained with FITC Anti-Human CD14 to clearly distinguish and gate populations of interest when analyzing the data using flow cytometry (Appendix A). The uptake of the HNPs into M_0_-MDMs was investigated in the same concentration range as previously stated (Appendix A). As shown in Figure 2B, a concentration-dependent uptake of the particles was revealed, whereby the overall internalization increased with an increasing concentration of the HNPs. This could be quantified by measuring the mean fluorescence intensity (MFI) of the internalized HNPs and comparing it to that of the PEG-PLGA nanoparticles. As demonstrated in Appendix A, there was a 1.5 to 3.3-fold increase in HNP uptake in M_0_-MDMs as compared to the PEG-PLGA nanoparticles, with the HNPs-NH_2_, HNP-COOH, and HNP-RGD exhibiting the highest degree of internalization at the highest concentration. The variation in the response can be attributed to the inherent heterogeneity of macrophages because they were freshly obtained from human donors.

CLSM was employed to track the internalization of HNPs and PEG-PLGA nanoparticles by M_0_-MDMs after following a 3 h incubation period at a concentration of 180 µg mL^−1^ (Figure 3 and Figure 4A). To facilitate efficient detection and tracking, the cell nuclei of the macrophages were stained with Hoechst 33342 dye (blue fluorescence), while the NLO was visualized as orange fluorescence. The results revealed that the HNP-COOH and HNP-RDG displayed the highest uptake as indicated by their increased fluorescence intensity. Conversely, CLSM demonstrated that the HNP-NH_2_, HNP-cRGD, and PEG-PLGA NPs exhibited considerably lower levels of uptake, corroborating the previous findings of the flow cytometry analysis. Additionally, the CLSM images revealed the fluorescence signal of the stained cellular membrane (red) surrounding the intracellular particles and the colocalization of the NLO signal (orange) with the lysosomal signal (green) (Figure 3). As the literature shows, endocytosis is the main process through which nanosized materials are taken up by cells to reach their respective targets [46,47]. Different characteristics of the particles, such as size, shape, and surface charge, can affect the internalization pathways [48,49]. Depending on these factors, the particles are internalized, leading to the formation of endosomes that engulf the particles, which then mature into lysosomes that facilitate the degradation in the material using lysosomal proteases [36]. Thus, as demonstrated by the results, the colocalization of the lysosomes and NLO signaling suggests a potential endocytic uptake pathway of the particles. Furthermore, the kinetics of the HNP uptake were investigated using CLSM, exemplified here by HNP-RGD. As shown in Figure 4B, the uptake of the particles was detectable after as early as 10 min of incubation, signaling a rapid and facilitated uptake by the macrophages.

### 3.3. Dual-Loaded PEG–Lipid–PLGA HNPs with Different Sizes and Functionalities

#### 3.3.1. Formulation of Dual-Loaded HNPs

Due to the enhanced uptake of HNP-COOH and HNP-RGD in M_0_-MDMs as well as their satisfactory stability in various media over weeks, further studies were conducted using these functionalized HNPs to evaluate their ability as carrier materials for the BRP-201 drug. Hence, dual encapsulation of the NLO dye and the BRP-201 anti-inflammatory agent in HNPs was subsequently performed (Appendix A). In addition to this, two additional HNP systems with mixed PEG–Lipids that carried a COOH or RGD functionality, i.e., HNP-COOH/RGD (1:1) and HNP-COOH/RGD (2:1), were prepared to investigate whether the amount of RGD functionality on the surface had an effect on the uptake in macrophages. Moreover, all functionalized HNPs and control NPs were formulated in two size ranges while aiming for average sizes below 150 nm and above 200 nm by using two different initial polymer concentrations (2.5 mg mL^−1^ and 25 mg mL^−1^) to investigate the passive targeting effect (Figure 5). Moving forward, small and large dual-loaded HNPs will be referred to as s-HNPs and l-HNPs, respectively.

The s-HNPs formulated with an initial polymer concentration of 2.5 mg mL^−1^ revealed sizes in the range of 140 nm, which was comparable to the size obtained with the pure NLO-encapsulated HNPs, whereby the l-HNPs prepared with 25 mg mL^−1^ revealed larger particle sizes in the range of 250 nm (Table 2 and Appendix A; Appendix A). However, the sizes given here only represent average values, and the particles did not actually differ by ~100 nm. Rather, the difference in size could be seen as a shift in the general size distribution. The exact size distribution of the individual particle systems and their standard deviations are displayed in the Appendix A (Appendix A). The PEG-PLGA NPs revealed slightly smaller average sizes for both concentrations (d_H_ = 135 nm and 170 nm).

All particles showed no change in size or PDI value over four weeks in water, indicating that the drug had no negative influence on the stability of the HNPs (Appendix A; Appendix A). The zeta potential of all loaded HNPs ranged from −35 to −41 mV in water and between −4 and −11 mV in NaCl, with the s- and l-PEG-PLGA NPs revealing lower zeta potentials in the range of −24 mV (Table 2 and Appendix A). SEM images of the particles displayed uniform and round-shaped particles for both the smaller and larger HNPs (Figure 6A). The mean size (d_N_) values were calculated from the SEM images via image processing and showed a d_N_ ranging from 77 nm to 99 nm for the s-HNPs and from 140 nm up to 194 nm for l-HNPs (Table 1; Appendix A).

The final BRP-201 and NLO loading was determined using UV–Vis measurements. LC values of 1.0% to 2.9% for BRP-201 and 0.06% to 0.11% for NLO were obtained for all HNPs (Table 2), with the l-HNPs encapsulating more drug and dye compared to the s-HNPs. This finding agrees with the literature, in which higher initial polymer concentrations and larger particle sizes led to increased loading [50]. Surprisingly, the PEG-PLGA NPs, especially the l-PEG-PLGA NPs, exhibited much higher LC values for BRP-201, even exceeding the amount of drug initially fed in some formulation batches. Consequently, in-depth SEM studies were performed to determine if free drug precipitates were present in the formulation, since similar observations were described in earlier studies when encapsulating BRP-201 in PLGA NPs [29]. The SEM analysis revealed that drug precipitates were present in all formulations to a small extent but were very prominent in the PEG-PLGA particles (Appendix A). Therefore, an NP batch of the l- and s-HNPs was filtered through a 0.8 µm cellulose acetate filter, and the LC was determined again. Moreover, the successful removal of the precipitates was investigated via additional SEM studies (Appendix A). When comparing the LC values before and after filtration, it can be seen that all values decreased substantially (by at least 32% and up to 70%). Similarly, the dye loading was determined after filtration, but the LC values changed only slightly compared to the drug (Appendix A; Appendix A). Hence, more BRP-201 precipitates were present in the formulations than NLO precipitates, which can be explained by the different amount that was initially used for encapsulation (3 wt% for BRP-201 and 0.1 wt% for NLO).

The HNPs were also examined for stability in PBS and acetate buffer systems. All particles showed consistent sizes and PDI values over three weeks in buffer (Appendix A; Appendix A). To investigate whether the drug loading and particle size had an effect on particle degradation, enzymatic degradation was investigated (Figure 6B). The results showed that the larger particles exhibited a longer degradation time compared to the smaller particles, as already known in the literature [51,52].

#### 3.3.2. HPLC Analysis of s-HNPs

The composition of formulated s-HNPs was monitored via HPLC after full dissolution of the lyophilized aliquots. In a previous study, drug-loaded PLGA NPs were analyzed quantitatively on a reversed-phase monolithic silica column, showing elution of the small drug molecule during an isocratic hold with the subsequent elution of PLGA after a steep gradient to 100% CH_3_CN in the mobile phase [53]. In the case of HNP analysis, two factors were additionally considered. Firstly, the presence of the lipids in the samples required a thermodynamically good solvent, i.e., CH_3_OH, for their dissolution and full elution from the column [54,55]. Secondly, a controlled pH was needed to adjust the charged state of the lipids for tuning their separation selectivity. Thus, the mobile phase consisted of (A) CH_3_CN, (B) aqueous buffer solution, and (C) CH_3_OH. The respective mobile phase composition was altered in a CH_3_OH gradient fashion (Appendix A).

The elution traces of s-PEG-PLGA NPs and s-HNPs with different functional lipids recorded via CAD are shown in Figure 7A. Peak groups between 10 and 15 min clearly distinguished lipids contained in s-HNPs when compared to mere s-PEG-PLGA NPs. Analysis of lipid and polymer standards separately (Figure 7B) allowed us to assign the signals to lipids with different functional groups, i.e., RGD and COOH, that did not show complete baseline separation. Simultaneous detection via DAD and FLD (Figure 7C, D) proved the presence of cargo, i.e., the BRP-201 drug and the NLO dye, in the final HNP-RGD formulation. Individual chromatograms of s-PEG-PLGA NPs and each s-HNP type (including the DAD and FLD traces) can be found in Appendix A. With the batch undergoing HPLC measurements, a brief study of quantification analysis was conducted (Appendix A; Appendix A) according to the previously developed protocol [53]. The amount of BRP-201 encapsulated in the s-PEG-PLGA NPs and s-HNPs (recorded via DAD) as well as the amount of PLGA (recorded via CAD) were determined using established calibration curves. The calibration was linear in the range of 5–40 µg mL^−1^ for BRP-201 (Appendix A). For the PLGA determination with a CAD, a double logarithmic plot of peak area against concentration was used that showed sufficient linearity in the concentration range of 0.1–5.0 mg mL^−1^ (Appendix A). The calculated LC values are presented in Appendix A. Although the quantification of the lipid components in the s-HNPs was not performed due to incomplete resolution between the DSPE-PEG-COOH and DSPE-PEG-RGD components (a resolution of only 0.94), an increased peak intensity (peak height) of DSPE-PEG-COOH in s-HNP-COOH/RGD (2:1) compared to s-HNP-COOH/RGD (1:1) indicated a larger amount of the component in the s-HNPs. Five repetitive injections of s-HNP-COOH (Appendix A) showed very low standard deviations of the PLGA and DSPE-PEG-COOH peak retention times, with a coefficient of variation of 0.07% and 0.02% and peak areas with a coefficient of variation of 0.29% and 0.61% (Appendix A), respectively. This made the developed method suitable for screening of future HNP formulations regarding qualitative and quantitative composition control.

#### 3.3.3. Uptake Studies of Dual-Loaded Particles in M_1_-MDMs

Differently functionalized s-HNPs and l-HNPs, as well as those with mixed functionality (loaded with both NLO and BRP-201), were tested for their uptake by M_1_-MDMs at a concentration of 100 µg mL^−1^ in water and PBS (1:1). As BRP-201 possesses anti-inflammatory properties, newly formulated NPs and HNPs were tested on the pro-inflammatory M_1_-MDMs to demonstrate their high capacity to generate pro-inflammatory leukotrienes and their enhanced phagocytic ability [34]. As evident in Figure 8, Appendix A, and Appendix A, varying the different functionalities demonstrated no particle uptake difference in the tested M_1_-MDMs among the HNPs themselves; however, when comparing them to the control PEG-PLGA NPs, a superior uptake of HNPs was detected. This was observed for both s-HNPs and l-HNPs. Additionally, l-HNPs and l-PEG-PLGA nanoparticles exhibited a higher uptake as compared to the smaller-sized particles. This could also be explained by the fact that larger particles could encapsulate more dye inside. This comes as no surprise, as the literature has already established that as the size of the particles increases, there is an enhancement in their uptake by macrophages [56]. These findings indicate that the mere existence of a lipid coating surrounding the polymeric core of the carrier system seems to improve the uptake of the particles, in contrast to the non-hybrid PEG-PLGA system [36,57,58].

An additional kinetic experiment was conducted to ensure that the observed cellular uptake rate expressed by MFI in the cells was not caused by free dye leakage from the HNPs. As seen in Figure 9 and Appendix A, at 15 min, a clear NLO signal could be detected inside the cells when treated with HNP-COOH and HNP-RGD. This could be distinguished at as early as 3 min; however, no NLO signal was visible when treated with PEG-PLGA nanoparticles and a free NLO dye solution prepared from DMSO and PBS/water. Thus, this confirmed that the dye remained within the particles as long as they were intact and not yet degraded. Moreover, the variation in the signaling among the HNPs also negated a dye leakage because otherwise, a similar pattern and intensity of NLO fluorescence inside the cells would have been detected. This has been demonstrated as well in another study [37].

#### 3.3.4. Investigations Using FLAP-Dependent Cell-Based Bioactivity Assays

In order to evaluate the ability of the HNPs to deliver the drug for sufficient suppression of FLAP-dependent 5-LOX product formation, neutrophils were preincubated for 15 min with BRP-201-loaded s-HNPs and s-PEG-PLGA NPs. Afterwards, the neutrophils were stimulated with 2.5 µM of calcium ionophore A23187 for 10 min to induce FLAP-dependent 5-LOX product formation. All particle systems decreased 5-LOX product formation by approx. 50%, indicating that the intracellular availability of BRP-201 was sufficient (Figure 10; Appendix A).

Furthermore, the particles were assessed with pro-inflammatory M_1_-MDMs that were preincubated with the HNPs for 15 min and then exposed to bacterial exotoxins from *Staphylococcus aureus (S. aureus*) (Figure 11). The *S. aureus*-conditioned medium (SACM) was obtained according to Jordan et al. and used at a concentration of 1% for 90 min to induce the biosynthesis of pro-inflammatory LMs, especially LTs in M_1_-MDMs [59]. In contrast to neutrophils, BRP-201-containing PEG-PLGA NPs only showed a minor inhibition in 5-LOX product formation in M_1_-MDMs, which could have been due to a reduced uptake rate by the M_1_-MDMs within the 15 min of incubation time (Figure 8). Also, HNPs formulated with DSPE-PEG-COOH/RGD (1:1) could just slightly reduce LTB_4_ formation, while HNP-COOH/RGD (2:1) and HNPs with COOH or RGD lipid functionalities impaired the FLAP-dependent LTB_4_ synthesis significantly (Figure 11A). RGD sequences bind to integrins and are involved in cell signaling [22], which may be an explanation for the inhibition effect these particles showed while PEG-PLGA NPs failed to suppress LTB_4_ synthesis. Interestingly, after using a shorter preincubation time of only 5 min, HNPs-COOH/RGD (2:1) could significantly inhibit LTB_4_ formation in M_1_-MDMs (Figure 11B). This experiment proves that a quick uptake is crucial for an intracellular pharmacological effect.

## 4. Conclusions

PEG–Lipid–PLGA HNPs with different surface functionalities, i.e., HNP-COOH, HNP-NH_2_, HNP-RGD, and HNP-cRGD, were formulated and loaded with an NLO dye or both the NLO dye and the anti-inflammatory FLAP inhibitor BRP-201 to study targeted transport to human MDMs and neutrophils. The suggested core–shell structure of the hybrid systems was confirmed via cryo-TEM, allowing the visualization and distinction of the lipid layer. The formulated HNP systems revealed stability over four weeks in water and three weeks in PBS and acetate buffer. The dye-loaded HNPs, particularly HNP-RGD and HNP-COOH, demonstrated a more efficient uptake by M_0_-MDMs compared to standard polymeric PEG-PLGA NPs, showing the influence of the functional group on the surface of the particles. Moreover, varying the overall mixed functionality did not result in a notable difference in uptake; however, varying the particle size did, whereby larger particles (d_H_ ~250 nm) were taken up in a more efficient manner than smaller ones (d_H_ ~140 nm). Composition of all the dual-loaded s-HNPs was monitored via HPLC. All the components of the s-HNPs, including BRP-201, NLO, lecithin, and PEG–Lipids with different functionalities, were found in the formulations. Simultaneous quantification of BRP-201 and PLGA was performed for one batch of s-HNPs and PEG-PLGA NPs. As low standard deviations of the component retention times were observed, the developed HPLC method can potentially be used for screening future HNP formulations with respect to qualitative and quantitative composition control. Additionally, s-HNPs, in particular the ones exhibiting the RGD function, were shown to successfully inhibit 5-LOX product formation and thus inflammation. This was achieved by delivering BRP-201 to the intracellular target site within less than 15 min. The most efficient inhibition was thereby induced by the HNP-COOH/RGD (2:1). These findings underscore the potential of functionalized HNPs as reliable and repeatable systems for enhancing drug delivery and improving their efficacy at specific target intracellular sites.

## Figures and Tables

**Figure 1 pharmaceutics-16-00187-f001:**
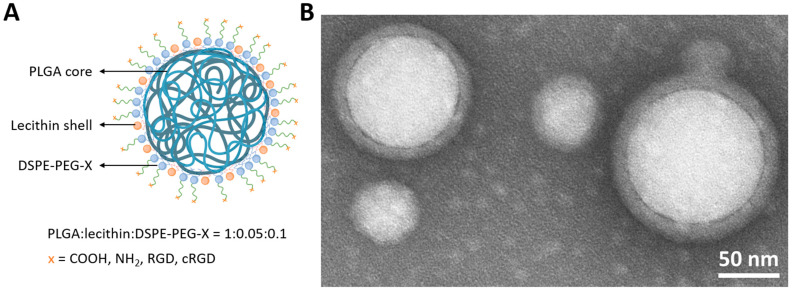
(**A**) Schematic representation of the structure of a hybrid nanoparticle (HNP) (created with BioRender.com, accessed on 2 October 2023) and (**B**) cryo-transmission electron microscopy (cryo-TEM) image of HNP-COOH negatively stained with UranyLess EM contrasting solution for 5 min (Science Services).

**Figure 2 pharmaceutics-16-00187-f002:**
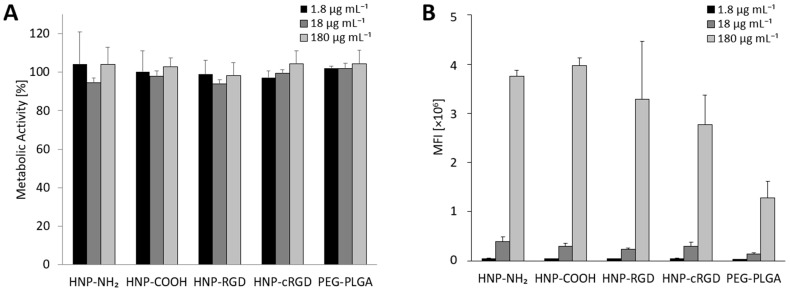
(**A**) Cytotoxicity of the HNPs and PEG-PLGA NPs tested in M_0_-MDMs at three different concentrations (180, 18, and 1.8 µg mL^−1^) at 24 h (n = 4 of one formulation batch). (**B**) Uptake of the HNPs and PEG-PLGA NPs in M_0_-MDMs at three different concentrations (1.8 and 18 µg mL^−1^ (n = 4) and 180 µg mL^−1^ (n = 2)). MFI = mean fluorescence intensity.

**Figure 3 pharmaceutics-16-00187-f003:**
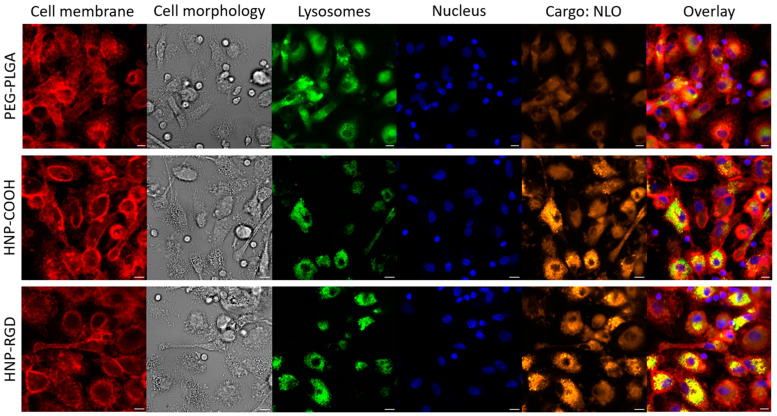
CLSM images of the M_0_-MDM uptake for PEG-PLGA NPs, HNP-COOH, and HNP-RGD at a concentration of 180 µg mL^−1^ at 3 h showcasing different organelle staining (red: CellMask™ Deep Red membrane stain (CMDR); green: LysoTracker™ Green DND-26; blue: Hoechst 33342; orange: NLO) (scale bar: 10 µm). The remaining CLSM images are shown in the Appendix A).

**Figure 4 pharmaceutics-16-00187-f004:**
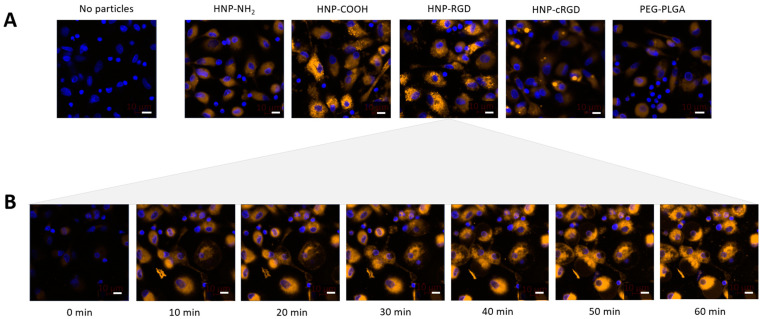
(**A**) CLSM images comparing M_0_-MDM uptake of PEG-PLGA NPs as well as the HNPs at a concentration of 180 µg mL^−1^ at 3 h (blue: Hoechst 33342; orange: NLO) (scale bar: 10 µm; magnification: 40×). (**B**) CLSM images of the M_0_-MDM uptake kinetics of HNP-RGD every 10 min over one hour at the same concentration.

**Figure 5 pharmaceutics-16-00187-f005:**
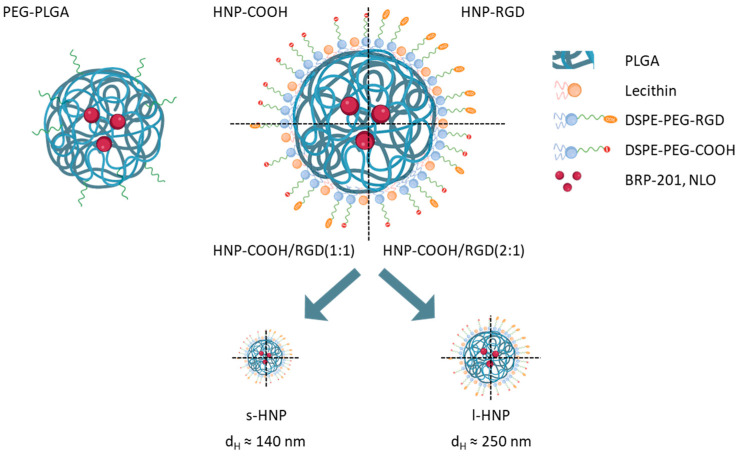
Schematic representation of PEG-PLGA particle and different functionalized HNPs formulated in two size ranges (s-HNPs and l-HNPs). Image created with BioRender.com, accessed on 2 October 2023.

**Figure 6 pharmaceutics-16-00187-f006:**
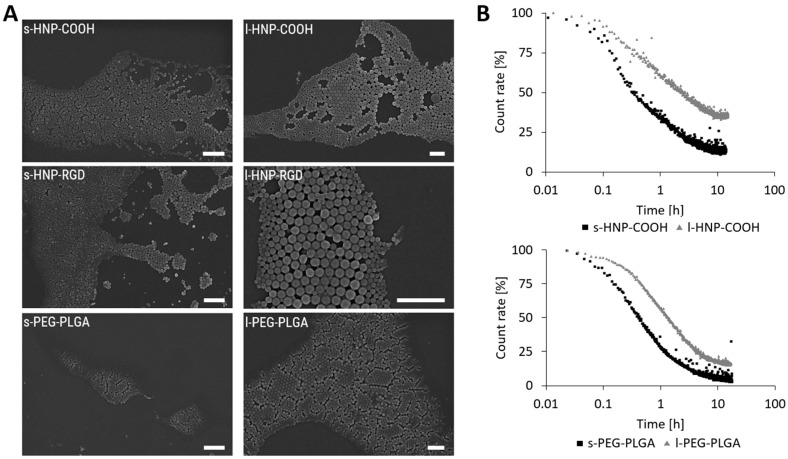
(**A**) SEM images of s- and l-PEG-PLGA NPs and s- and l-HNPs (scale bar = 1 µM). (**B**) Enzymatic degradation of the s- and l-PEG-PLGA NPs and s- and l-HNPs-COOH. Particles were mixed at a 1:2 mass ratio with proteinase K. Degradation was observed by monitoring the count rate and size via DLS. n = 1.

**Figure 7 pharmaceutics-16-00187-f007:**
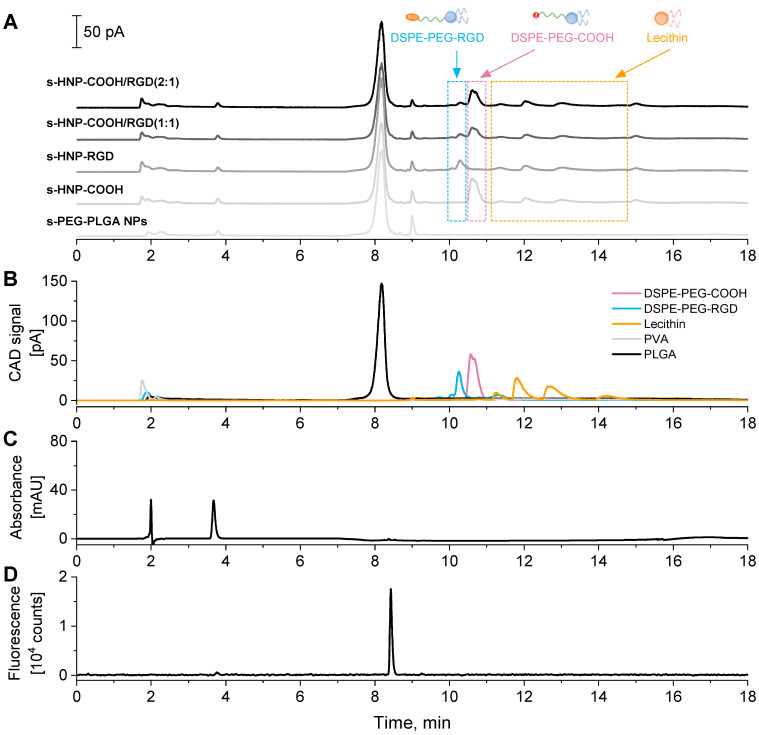
(**A**) Elugrams of dual-loaded (with the BRP-201 drug and the NLO dye) s-PEG-PLGA NPs and s-HNPs with different functional lipids. (**B**) Elugrams of PLGA, DSPE-PEG-COOH, DSPE-PEG-RGD, lecithin, and PVA standards. For simplicity of interpretation, the signal intensities of DSPE-PEG-COOH, DSPE-PEG-RGD, lecithin, and PVA are multiplied by a factor of 0.25. (**C**) Elugram of s-HNP-RGD recorded via DAD at 312 nm. The peak at 3.7 min refers to BRP-201. (**D**) Elugram of s-HNP-RGD recorded via FLD (λ_ex_ = 555 nm, λ_em_ = 592 nm). The peak at 8.4 min refers to NLO. Measurement conditions: flow rate 0.75 mL min^−1^, CH_3_CN/water with 10 mM of ammonium acetate (pH 5.5)/CH_3_OH with 10 mM ammonium acetate. The gradient elution conditions can be found in Appendix A.

**Figure 8 pharmaceutics-16-00187-f008:**
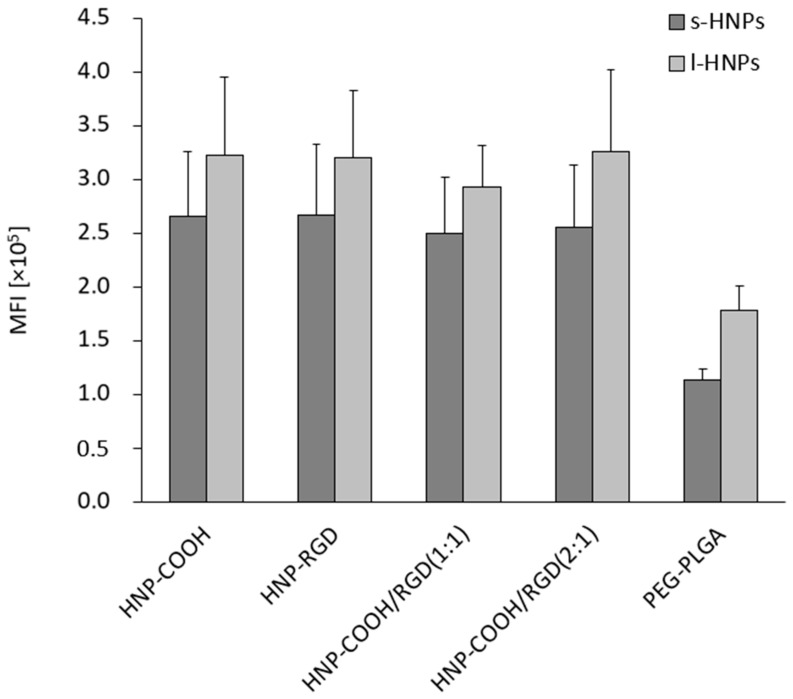
Uptake of the s- and l-HNPs as well as s- and l-PEG-PLGA NPs in M_1_-MDMs at 100 µg mL^−1^ (n = 2). MFI = mean fluorescence intensity. No statistical significance was established due to a limited sample size.

**Figure 9 pharmaceutics-16-00187-f009:**
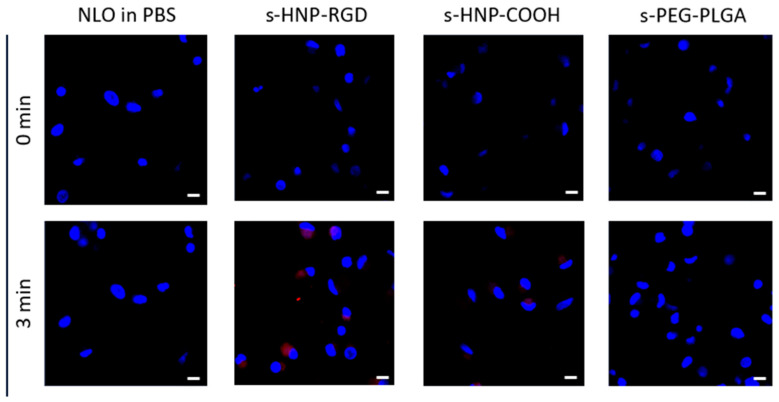
M_1_-MDM uptake kinetics of the s-HNP-COOH and s-HNP-RGD as compared to s-PEG-PLGA NP at a concentration of 100 µg mL^−1^ and free NLO in DMSO at 0.06 µg mL^−1^ (representative of the %LC of the HNP) using CLSM at 0 and 3 min (scale bar: 10 µm; magnification: 40×).

**Figure 10 pharmaceutics-16-00187-f010:**
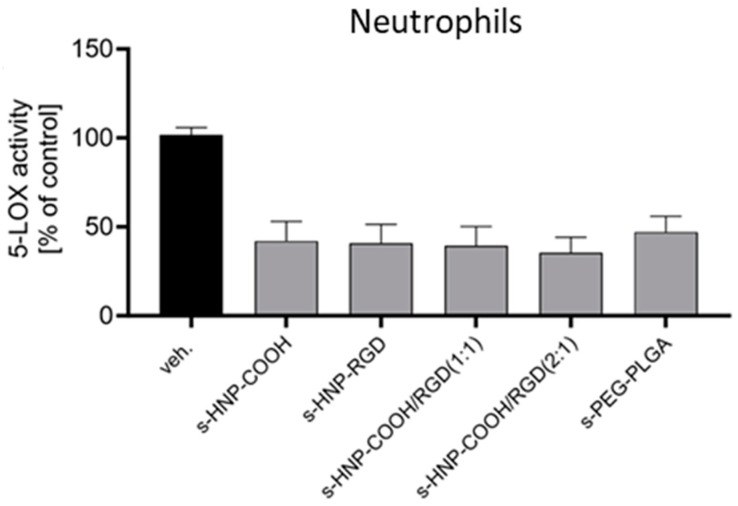
Suppression of 5-LOX product formation by encapsulated BRP-201 in s-HNPs and s-PEG-PLGA NPs or PBS (vehicle (veh.) control) in neutrophils. Neutrophils were preincubated with particles (encapsulated with BRP-201 at a concentration of 0.3 µM) for 15 min at 37 °C and then stimulated with 2.5 µM of A23187. After 10 min, the reaction was stopped, and 5-LOX products were extracted via solid-phase extraction (SPE) and analyzed via HPLC. Values are given as 5-LOX products (LTB_4_, trans-LTB_4_, epi-trans-LTB_4_, and 5-HETE) in percentage (%) of control.

**Figure 11 pharmaceutics-16-00187-f011:**
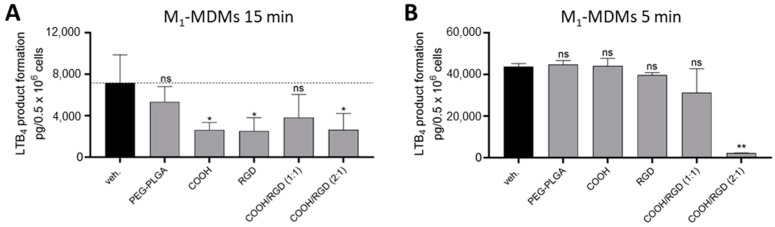
Inhibition in LTB4 levels by encapsulated BRP-201 in M_1_-MDMs. (**A**) M_1_-MDMs were preincubated with PBS (veh. control) or HNPs and NPs (encapsulated with BRP-201 at a 0.3 µM final concentration) for 15 min at 37 °C and then stimulated with SACM (1%). The reaction was stopped after 90 min with methanol, and LMs were extracted via solid-phase extraction (SPE) and analyzed via UPLC-MS-MS. (**B**) M_1_-MDMs were preincubated with PBS (veh. control) or particles (encapsulated with BRP-201 at a 0.3 µM final concentration) for 5 min at 37 °C and then stimulated with 2.5 µM of A23187. The reaction was stopped after 10 min with methanol, and LMs were extracted via SPE and analyzed via UPLC-MS-MS. Values are given as pg/1 × 10^6^ cells. Statistical analysis was assessed using a ratio paired *t*-test (n = 3). ns = Not significant; * *p* < 0.05, ** *p* < 0.01.

**Table 1 pharmaceutics-16-00187-t001:** Properties of the NLO-loaded HNPs prepared with different functionalized PEG–Lipids as well as the control PEG-PLGA NPs (n = 5 replicates).

Sample	d_H_ (nm) ± SD	PDI ± SD	ζ (mV) ± SD	Yield ± SD (%) n = 4	LC (%) ± SD	d_N_ (nm)
HNP-NH_2_	147 ± 4	0.14 ± 0.05	−20 ± 6	47 ± 8	0.06 ± 0.00	87
HNP-COOH	150 ± 20	0.15 ± 0.10	−35 ± 12	67 ± 13	0.07 ± 0.01	90
HNP-RGD	150 ± 20	0.08 ± 0.01	−35 ± 5	72 ± 7	0.06 ± 0.01	98
HNP-cRGD	137 ± 3	0.07 ± 0.02	−29 ± 5	41 ± 11	0.07 ± 0.01	96
PEG-PLGA	122 ± 4	0.07 ± 0.01	−14 ± 15	64 ± 9	0.05 ± 0.00	72

All HNPs contained lecithin next to the PEG–Lipid conjugate. All particle systems were loaded with NLO dye. d_H_ = Hydrodynamic diameter obtained via DLS, PDI = polydispersity index, ζ = zeta potential, LC = loading capacity, SD = standard deviation between five independent formulations, d_N_ = number-weighted size obtained via the analysis of SEM images. Yield determined after filtration through a 0.8 µM cellulose acetate filter.

**Table 2 pharmaceutics-16-00187-t002:** Properties of HNPs and NPs loaded with BRP-201 and NLO as cargo (n = three replicates).

Sample	d_H_ (nm) ± SD	PDI ± SD	ζ (mV) ± SD	LC_BRP-201_ (%) ± SD	LC_NLO_ (%) ± SD	d_N_ (nm)
s-HNP-COOH	153 ± 10	0.32 ± 0.04	−40 ± 6	1.24 ± 0.22	0.07 ± 0.01	94
s-HNP-RGD	143 ± 5	0.27 ± 0.05	−36 ± 4	1.00 ± 0.07	0.06 ± 0.01	99
s-HNP-COOH/RGD (1:1)	139 ± 4	0.24 ± 0.04	−41 ± 7	1.07 ± 0.35	0.07 ± 0.01	89
s-HNP-COOH/RGD (2:1)	142 ± 5	0.25 ± 0.03	−35 ± 2	1.15 ± 0.21	0.07 ± 0.01	77
s-PEG-PLGA	137 ± 33	0.23 ± 0.09	−24 ± 6	1.33 ± 0.28	0.06 ± 0.01	84
l-HNP-COOH	249 ± 19	0.29 ± 0.03	−41 ± 0	1.37 ± 0.29	0.08 ± 0.00	150
l-HNP-RGD	234 ± 16	0.25 ± 0.05	−34 ± 7	1.38 ± 0.40	0.08 ± 0.00	162
l-HNP-COOH/RGD (1:1)	256 ± 25	0.28 ± 0.03	−35 ± 2	1.70 ± 0.11	0.08 ± 0.00	194
l-HNP-COOH/RGD (2:1)	252 ± 19	0.27 ± 0.32	−35 ± 7	1.38 ± 0.22	0.07 ± 0.01	140
PEG-PLGA	174 ± 6	0.21 ± 0.04	−24 ± 0	2.89 ± 0.33	0.11 ± 0.01	143

All HNPs contained lecithin next to the functional PEG–Lipid (indicated by the sample name). All particle systems were loaded with the NLO dye and the BRP-201 drug. d_H_ = Hydrodynamic diameter obtained via DLS, PDI = polydispersity index, ζ = zeta potential, LC = loading capacity, SD = standard deviation between measurements of three independent formulations, d_N_ = number-weighted size obtained via the analysis of SEM images. Yield determined after filtration through a 0.8 µM cellulose acetate filter.

## Data Availability

Data are contained within the article and Appendix A.

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
