# Peer review of "PEG–Lipid–PLGA Hybrid Particles for Targeted Delivery of Anti-Inflammatory Drugs"

_pharmaceutics, 2024, doi:10.3390/pharmaceutics16020187_

Round 1
Reviewer 1 Report
Comments and Suggestions for Authors
The manuscript entitled “PEG-Lipid-PLGA hybrid particles for targeted delivery of anti-inflammatory drugs” contains some interesting findings, and the authors mentioned as follows:
Hybrid nanoparticles (HNPs), composed of a PLGA core and a lipid shell incorporating PEG-lipid conjugates, were designed with various functionalities (RGD, cRGD, NH2, COOH) to create diverse drug delivery systems. Loaded with neutral lipid orange dye, these HNPs were examined for uptake in human monocyte-derived macrophages (MDMs) using FC and CLSM. Additionally, HNPs loaded with the anti-inflammatory drug BRP-201 were prepared in two size ranges (dH ~140 nm and dH ~250 nm) and evaluated for particle characteristics, stability, and degradation. Analysis included DLS, SEM, HPLC, SEC, and UV/Vis spectroscopy. Uptake studies indicated superior MDM uptake for HNP-COOH and HNP-RGD, with HNP-COOH/RGD(2:1) demonstrating the highest inhibition of 5-LOX product formation.
However, the manuscript needs many more characterizations and data to improve the results and discussion. I thus recommend the paper be reconsidered after major revisions.
The reviewer has the following comments
Abstract: The abstract is excessively lengthy, making it challenging to follow. Consider a concise rewrite that focuses directly on the original findings of the research, avoiding overly generic statements. Specifically, refrain from discussing instrument techniques.
Statistical Analysis: It is recommended to include a dedicated section on statistical analysis in the revised manuscript.
Experimental Data: Include NMR analysis, TGA, and DSC curves of NPs in the experimental data section.
Introduction: Revise the introduction to clearly highlight the scientific problems addressed by the research. While various biocompatible systems exist, such as hydrogels, nanofibers, nanofilms, nanocomposites, etc., justify the choice of NPs over other systems. Reference relevant articles to emphasize the familiarity and advantages of NPs.
In-depth Introduction: For a more comprehensive introduction, refer to recent articles by EV. Barrera, Seungpyo Hong, Ramiro Manuel, R.J. Linhardt, and N. Mamidi. Incorporate insights from these works to provide a thorough overview of nano-formulations. A deeper exploration of this research domain will enhance the manuscript's quality and engage readers.
Cell Viability Measurement: Consider extending the cell viability measurement beyond 24 hours for a more comprehensive assessment.
Conclusions: Modify the conclusions based on the revised data, incorporating more quantitative information to strengthen the findings. Provide a clear and data-driven summary of the research outcomes.
Author Response
Reviewer #1: The manuscript entitled “PEG-Lipid-PLGA hybrid particles for targeted delivery of anti-inflammatory drugs” contains some interesting findings, and the authors mentioned as follows:
Hybrid nanoparticles (HNPs), composed of a PLGA core and a lipid shell incorporating PEG-lipid conjugates, were designed with various functionalities (RGD, cRGD, NH2, COOH) to create diverse drug delivery systems. Loaded with neutral lipid orange dye, these HNPs were examined for uptake in human monocyte-derived macrophages (MDMs) using FC and CLSM. Additionally, HNPs loaded with the anti-inflammatory drug BRP-201 were prepared in two size ranges (dH ~140 nm and dH ~250 nm) and evaluated for particle characteristics, stability, and degradation. Analysis included DLS, SEM, HPLC, SEC, and UV/Vis spectroscopy. Uptake studies indicated superior MDM uptake for HNP-COOH and HNP-RGD, with HNP-COOH/RGD(2:1) demonstrating the highest inhibition of 5-LOX product formation.
However, the manuscript needs many more characterizations and data to improve the results and discussion. I thus recommend the paper be reconsidered after major revisions.
We would like to thank the reviewer for the critical and valuable comments. Please find our answers and changes in the following.
The reviewer has the following comments:
Abstract: The abstract is excessively lengthy, making it challenging to follow. Consider a concise rewrite that focuses directly on the original findings of the research, avoiding overly generic statements. Specifically, refrain from discussing instrument techniques.
We agree that the abstract was too long and too much details were mentioned. We shortened the abstract as follows:
“Hybrid nanoparticles (HNPs) were designed by combining a PLGA core with a lipid shell that incorporates PEG-lipid conjugates with various functionalities (-RGD, -cRGD, -NH2, -COOH) to create targeted drug delivery systems. Loaded with a neutral lipid orange dye, the HNPs were extensively characterized using various techniques and investigated for their uptake in human monocyte-derived macrophages (MDMs) using FC and CLSM. Moreover, the best performing HNPs (i.e., HNP-COOH and HNP-RGD as well as HNP-RGD/COOH mixed) were loaded with the anti-inflammatory drug BRP-201 and prepared in two size ranges (dH ~140 nm and dH ~250 nm). The HNPs were examined further for their stability, degradation, MDM uptake as well as drug delivery efficiency by studying the inhibition of 5-lipoxygenase (5-LOX) product formation, whereby HNP-COOH and HNP-RGD exhibited both superior uptake, and the HNP-COOH/RGD(2:1) displayed the highest inhibition.”
Statistical Analysis: It is recommended to include a dedicated section on statistical analysis in the revised manuscript.
Thank you for pointing this out. Statistical analyses were performed on the bioactivity assays results, and as advised were added in the methods section.
“Statistical Analysis
Results are given as mean ± standard error of each independent experiment, where n represents the indicated number from different donors. Statistical analysis and graphs were made by using GraphPad Prism 9 software (San Diego, CA, USA). Statistical analysis was assessed using a ratio paired t-test with a p value below 0.05 indicating a significant result.”
Experimental Data: Include NMR analysis, TGA, and DSC curves of NPs in the experimental data section.
We thank the reviewer for this suggestion. We understand that additional experiments with many more techniques would be of high interest. However, would like to state here that we have performed already a comprehensive analysis of the particles and their properties as can be seen by the amount of data included in the Main and the Supporting Information. We also believe that we described the data in a manner that it is understandable for the reader.
We are, however, aware that these techniques are very valuable to gain insight into the deeper structure of the particles. Unfortunately, in depth DSC and NMR investigations require a lot of material and time and hence cannot be performed with an expected revision within 10 days. Also, DSC and TGA measurements has been already performed to characterize several PLGA-DSPE-PEG systems, and since we based our formulation protocols on these previously described hybrid systems, we believe there is no need to repeat the DSC analyses for our PLGA-DSPE-PEG particles. Kindly find below a few references for your review that confirm our claims:
- F. Yu, M. Ao, X. Zheng, N. Li, J. Xia, Y. Li, D. Li, Z. Hou, Z. Qi, X. D. Chen, Drug Delivery 2017, 24, 825-833.
- N. Tahir, A. Madni, A. Correia, M. Rehman, V. Balasubramanian, M. M. Khan, H. A. Santos, Int. J. Nanomed. 2019, 14, 4961-4974.
- H. A. Mohammad, M. M. Ghareeb, M. Akrami, A. S. Sahib, J. Complementary Med. Res. 2020, 11, 204-214.
Moreover, we also agree that NMR analysis is a useful and informative technique for this type of system, e.g. to confirm the presence of the PEG shell and the targeting function on the outside of the particles. However, as we performed SEC and HPLC measurements to determine the composition of the particles as well as accompanied the characterization with complementary cryo-TEM imaging to confirm the core-shell structure of the particles, NMR analysis would not provide any additional valuable information that can strengthen our characterization. Additionally, as demonstrated by in vitro testing, particles of same composition but different functionalities on the surface exhibit different uptake behavior, confirming the structure with the targeting groups presented on the surface.
Nonetheless, further studies are planned to focus on the best performer, and we would consider to include NMR analysis as suggested by the reviewer, if appropriate.
Introduction: Revise the introduction to clearly highlight the scientific problems addressed by the research. While various biocompatible systems exist, such as hydrogels, nanofibers, nanofilms, nanocomposites, etc., justify the choice of NPs over other systems. Reference relevant articles to emphasize the familiarity and advantages of NPs.
In-depth Introduction: For a more comprehensive introduction, refer to recent articles by EV. Barrera, Seungpyo Hong, Ramiro Manuel, R.J. Linhardt, and N. Mamidi. Incorporate insights from these works to provide a thorough overview of nano-formulations. A deeper exploration of this research domain will enhance the manuscript's quality and engage readers.
The manuscript introduces the aspect of drug delivery and promising materials, such as polymers and lipids, and then focuses on the hybrid nanoparticles (HNPs) as delivery systems. We believe that it is of utmost importance that the HNPs are elaborated on in detail so that the reader clearly understands why we are using polymers, lipids and core-shell hybrid systems. Moreover, this manuscript is not a review, and the focus is not to provide an overview of drug delivery systems. As such, listing, describing and discussing the advantages and disadvantages of other various systems, such as nanofibers, would result in an unnecessary and inappropriate extension of the introduction and divert the reader's attention from the primary focus, i.e. hybrid nanoparticle systems.
We did, however, kindly added a sentence about drug delivery systems in general to the introduction and cited accordingly, as well as a citation of one of Prof. Hong works, which provides an insight into the targeting capabilities of NPs, in addition to the novel review by Jain et al., which delivers a comprehensive overview of Lipid-Polymer-Hybrid NPs and nicely highlights the pros and cons of these carriers.
References:
1. N. Mamidi, R. M. Velasco Delgadillo, E. V. Barrera, in Pharmaceuticals, Vol. 14, 2021.
2. X. Hu, S. Liu, G. Zhou, Y. Huang, Z. Xie, X. Jing, J. Control. Release. 2014, 185, 12-21.
3. V. H. DeLeon, T. D. Nguyen, M. Nar, N. A. D'Souza, T. D. Golden, Mater. Chem. Phys. 2012, 132, 409-415.
4. F. X. Gu, R. Karnik, A. Z. Wang, F. Alexis, E. Levy-Nissenbaum, S. Hong, R. S. Langer, O. C. Farokhzad, Nano Today 2007, 2, 14-21.
5. S. Jain, M. Kumar, P. Kumar, J. Verma, J. M. Rosenholm, K. K. Bansal, A. Vaidya, J. Funct. Biomater. 2023, 14, 437.
Cell Viability Measurement: Consider extending the cell viability measurement beyond 24 hours for a more comprehensive assessment.
The choice of a 24-hour timeframe was deliberately selected taking into consideration that it is an initial assessment of these particle systems, and that as the study progressed, it was evident that the uptake was prominent within a 3-hour frame, so extending the cell viability measurement beyond that is unnecessary in our opinion.
We would, however, definitely take that into consideration for our follow-up studies where we are aiming for the optimization of the in vitro testing as well as expanding into an in vivo model for the investigation of the best performer.
Conclusions: Modify the conclusions based on the revised data, incorporating more quantitative information to strengthen the findings. Provide a clear and data-driven summary of the research outcomes.
Thank you for your comment. As no additional quantitative data was added to the manuscript, the conclusion was modified taking into consideration the provided data and additional outcomes were reiterated to highlight the extensive characterization performed.
“PEG-Lipid-PLGA HNPs with different surface functionalities, i.e., HNP-COOH, HNP-NH2, HNP-RGD, HNP-cRGD, were formulated and loaded with the dye NLO or both the dye NLO and the anti-inflammatory FLAP inhibitor BRP-201 to study targeted transport to human MDMs and neutrophils. The suggested core-shell structure of the hybrid systems was confirmed by cryo-TEM, allowing the visualization and distinction of the lipid layer. Formulated HNP systems revealed stability over four weeks in water and three weeks in PBS and acetate buffer. The dye-loaded HNPs, particularly HNP-RGD and HNP-COOH, have demonstrated a more efficient uptake by M0-MDMs compared to standard polymeric PEG-PLGA NPs, showing influence of the functional group on the surface of the particles. Moreover, varying the overall mixed functionality did not result in a notable difference in uptake; however, varying the particle size did, whereby larger particles (dH ~250 nm) were taken up in a more efficient manner than smaller ones (dH ~140 nm). Composition of all the dual loaded s-HNPs was monitored by HPLC. All the components of the s-HNPs including BRP-201, NLO, lecithin and PEG-Lipids with different functionalities were found in the formulations. Simultaneous quantification of BRP-201 and PLGA was performed for one batch of s-HNPs and PEG-PLGA NPs. As low standard deviations of the component retention times were observed, the developed HPLC method potentially can be used for screening future HNP formulations with regard to qualitative and quantitative composition control.Additionally, s-HNPs, in particular the ones exhibiting the RGD function, have been shown to successfully inhibit 5-LOX product formation and, thus, inflammation. This is achieved by delivering BRP-201 to the intracellular target site in within less than 15 min. The most efficient inhibition was, thereby, induced by the HNP-COOH/RGD(2:1). These findings underscore the potential of functionalized HNPs as reliable and reproducible systems for enhancing drug delivery and improving their efficacy at specific target intracellular sites.”
Reviewer 2 Report
Comments and Suggestions for Authors
Congratulations to the authors, there is a very interesting work.
I just recommend increase the number of references to discuss all the results in general.
It would have been interesting to know if the release of the drug was modified with the functionalizations carried
Materials section: The authors doesn't include the information related with the supplier of the BRP-201, Also I recommend the inclusion of the molecular formula and chemical structure of the BRP-201 molecule.
Methods section: The authors should provide a more detailed description of the speed stirring conditions in the nanoprecipation technique for nanoparticles synthesis.
Line 477 include references for the particles characteristics
Line 678 mark an reference error citation
Author Response
Reviewer #2: Congratulations to the authors, there is a very interesting work.
I just recommend increase the number of references to discuss all the results in general.
Thank you for pointing this out. Many references have been included as marked in yellow in the manuscript.
It would have been interesting to know if the release of the drug was modified with the functionalizations carried
Thank you for pointing out this valuable comment. Due to the fact that the hydrophobic cargo crashes out once released in the aqueous phase, it is not possible to quantify the accurate release profile. Thus, we investigated the degradation profile of HNPs and the release of the active substance alone was not investigated. The release of the active ingredient, however, is an inevitable consequence of the complete degradation of the particle. In our study, the investigation of the degradation of the HNPs showed no difference between differently modified particles, so the release of the active substance should be the same.
Materials section: The authors doesn't include the information related with the supplier of the BRP-201, Also I recommend the inclusion of the molecular formula and chemical structure of the BRP-201 molecule.
We synthesize the active ingredients in-house with an already published protocol. This information is important and has been forgotten to be included in the material section. We have rectified this omission. In the introduction, we have also referred again to the structure of the active ingredient in our supporting information.
“BRP-201 was synthesized according to a published procedure.[30]”
The present study aimed to employ HNPs for the targeted transport to human monocyte-derived macrophages (MDMs) and neutrophils for an enhanced delivery of the anti-inflammatory drug BRP‑201 (5-{1-[(2-chlorophenyl)methyl]-2-{1-[4-(2-methylpropyl)-phenyl]ethyl}-1H-benzimidazole-5-yl}-2,3-dihydro-1,3,4-oxadiazole-2-thione) (Structure shown in Figure S5).
Methods section: The authors should provide a more detailed description of the speed stirring conditions in the nanoprecipitation technique for nanoparticles synthesis.
Thanks for pointing this out, it was indeed omitted. The particles were stirred at 800 rpm at room temperature. This information was added in the methods section.
“The mixture was stirred continuously at 800 rpm at room temperature under exclusion of light, allowing the organic solvent to evaporate overnight.”
Line 477 include references for the particles characteristics
We thank the reviewer for the comment. The following references were added:
- J. J. Rennick, A. P. R. Johnston, R. G. Parton, Nat. Nanotechnol. 2021, 16, 266-276.
- T.-G. Iversen, T. Skotland, K. Sandvig, Nano Today 2011, 6, 176-185.
- A. Vollrath, A. Schallon, C. Pietsch, S. Schubert, T. Nomoto, Y. Matsumoto, K. Kataoka, U. S. Schubert, Soft Matter 2013, 9, 99-108.
- A. Sahin, G. Esendagli, F. Yerlikaya, S. Caban-Toktas, D. Yoyen-Ermis, U. Horzum, Y. Aktas, M. Khan, P. Couvreur, Y. Capan, Artif. Cells Nanomed. Biotechnol. 2017, 45, 1657-1664.
Line 678 mark an reference error citation
Thank you for your comment. This error was rectified.
Reviewer 3 Report
Comments and Suggestions for Authors
The study was well-designed and well-executed. Here are a few minor suggestions for potential improvement.
1. Please insert the appropriate citation in line 396
2. Please fix the content in lines 475 and 678.
3. How did the authors confirm the transition of M0 GM-CSF MDMs to M1-MDMs after using LPS and IFN treatment? Were the inflammatory cytokines or M1-MDMs-related surface makers assessed for confirmation?
Comments on the Quality of English LanguageOverall, the writing was clear and comprehensible. The authors have provided sufficient information, enhancing the reproducibility of the experiment.
Author Response
Reviewer #3: The study was well-designed and well-executed. Here are a few minor suggestions for potential improvement.
We would like to thank the reviewer for these suggestions.
1. Please insert the appropriate citation in line 396
The reference was added accordingly.
2. Please fix the content in lines 475 and 678.
Thank you for pointing that out. We corrected the errors.
3. How did the authors confirm the transition of M0 GM-CSF MDMs to M1-MDMs after using LPS and IFN treatment? Were the inflammatory cytokines or M1-MDMs-related surface makers assessed for confirmation?
The polarization follows an established protocol. Surface markers were for example assessed in our lab by Werner et al. (doi: 10.1096/fj.201802509R.) and the same protocol was used.
Comments on the Quality of English Language
Overall, the writing was clear and comprehensible. The authors have provided sufficient information, enhancing the reproducibility of the experiment.
Thank you for your kind words!
Round 2
Reviewer 1 Report
Comments and Suggestions for Authors
No more comment!